# Natremia Significantly Influences the Clinical Outcomes in Patients with Severe Traumatic Brain Injury

**DOI:** 10.3390/diagnostics15020125

**Published:** 2025-01-07

**Authors:** Bharti Sharma, Winston Jiang, Munirah M. Hasan, George Agriantonis, Navin D. Bhatia, Zahra Shafaee, Kate Twelker, Jennifer Whittington

**Affiliations:** 1Elmhurst Hospital Center, Trauma Unit, Department of Surgery, NYC Health & Hospitals, New York, NY 11373, USA; jiangw3@nychhc.org (W.J.); hassanm14@nychhc.org (M.M.H.); agriantg@nychhc.org (G.A.); bhatian1@nychhc.org (N.D.B.); shafaeez1@nychhc.org (Z.S.); twelkerk1@nychhc.org (K.T.); harrisj20@nychhc.org (J.W.); 2Trauma Unit, Department of Surgery, Icahn School of Medicine at Mount Sinai Hospital, New York, NY 10029, USA

**Keywords:** sodium levels, hypernatremia, traumatic brain injury, injury severity score, severe trauma

## Abstract

**Objective**: Fluctuations in sodium levels (SLs) may increase mortality, severity, and prolonged length of stay (LOS) in critically ill patients. We aim to study the effect of SL on various clinical outcomes in patients with severe traumatic brain injury (TBI). **Methods**: This is a single-center, retrospective study of patients with severe TBI from 1 January 2020 to 31 December 2023, inclusive. Patients were identified using Abbreviated Injury Severity (AIS) scores and International Classification of Diseases (ICD) injury descriptions. **Result**: Variations in hospital (H) admission SLs were statistically significant across four age ranges (pediatric, young adult, older adults, and elderly). Intensive care unit (ICU) admission, H discharge, and death also showed significance. A statistical difference was noted in ICU discharge levels while comparing blunt versus penetrating injury. We found statistically significant differences in SLs at H admission, ICU admission, and ICU discharge when compared to the Injury Severity Score (ISS) and the Glasgow Coma Scale (GCS) at admission. A linear regression analysis revealed a statistically significant positive correlation between ICU admission SLs and ISS. We discovered statistically significant differences when comparing ICU admission levels to H LOS, ventilator days, and mortality. **Conclusions**: SL upon ICU admission is correlated with ISS, GCS, and mortality rates. The elevated admission SL was linked to adverse hospital outcomes, including prolonged LOS at the H, ICU, and mechanical ventilation. Moreover, variability in serum SLs is independently associated with mortality throughout the hospital stay, irrespective of the absolute serum sodium concentration.

## 1. Introduction

Traumatic brain injury (TBI) is a major public health issue, impacting around 1.4 million people in the United States each year, resulting in 235,000 hospitalizations and 50,000 deaths annually [1]. More than 40% of survivors experience long-term disabilities, underscoring the profound impact on patients and healthcare systems [1]. Despite advancements in neurocritical care, nearly one-third of patients with severe TBIs die, and less than half achieve a favorable neurological outcome [1]. Sodium imbalances, such as hyponatremia and hypernatremia, are prevalent in TBIs and are linked to increased mortality and poorer outcomes.

Human serum sodium levels (SLs) are tightly regulated, but individuals with traumatic brain injury (TBI) have a heightened risk of dysnatremia [1]. This is due to factors such as hyperosmolar treatment, diabetes insipidus, the use of fluids with high sodium content, and the choice of nutritional preparation by the provider. Some factors related to trauma include posterior pituitary dysfunction and inappropriate water retention [2,3].

In TBI patients, the prevalence of hyponatremia was reported to range from approximately 15% to 55% [4,5,6], while the incidence ranges from 9.6% to 51% [7]. Hypernatremia affects about 30% to 50% of individuals, with an incidence ranging from 16% to 40% [1]. Both conditions were individually linked to higher mortality rates and poorer outcomes [5].

Fluctuations in serum sodium levels, as well as overall sodium concentrations, can have significant pathophysiological effects. Therefore, it is crucial to manage serum sodium, especially for patients suffering from elevated intracranial pressure. Beyond the context of TBI, recent studies have highlighted the importance of monitoring changes in serum sodium over time to better understand its relationship with patient outcomes. Evidence suggests that sodium level fluctuations contribute to increased mortality, independent of the actual sodium level [8,9].

This study investigates how variations in SLs and the overall SL affect clinical outcomes in TBI patients. Given the global prevalence of TBI and its substantial impact on physical and cognitive function, identifying early and reliable prognostic factors is essential to guide clinical decision-making, the evaluation of quality of treatment, and medical resource allocation.

Given the high incidence and prevalence of sodium dysregulation in TBI patients and its significant implications for management and outcomes, identifying early and reliable prognostic factors is essential. This knowledge can guide clinical decision-making, improve the quality of treatment, and aid in the efficient allocation of medical resources. This study investigates how variations in SLs and overall sodium variability affect clinical outcomes in TBI patients, emphasizing the incidence and prevalence of sodium dysregulation and its relevance in patient management and outcomes.

We hypothesize that extremes in SLs and their fluctuations reflect disease severity and independently influence the prognosis of patients with TBIs. Changes in SLs can serve as indicators of disease severity and may have an independent effect on the prognosis of critically ill patients. Our primary objective is to investigate the impact of SLs on clinical outcomes in TBI patients. The clinical outcomes evaluated include adverse hospital events, mortality rates, length of hospital stay (LOS), length of stay in the intensive care unit (ICU), and duration of mechanical ventilation. This study aims to highlight sodium’s potential as a biomarker and a target for optimizing patient care.

## 2. Methods

### 2.1. Study Population

This is a single-center, retrospective review conducted at a level 1 trauma center verified by the American College of Surgeons in Queens, New York City. We included all patients who presented with a severe TBI between 1 January 2020 and 31 December 2023, inclusive. All patients with an Abbreviated Injury Severity (AIS) score of 3 or higher were included. Patients who were discharged or transferred and had SLs collected were also included in the analysis. We excluded patients who had a COVID-19 infection at the time of their injury, who died or were discharged within 24 h of their initial injury, and who had non-severe and minor injuries. We found a total of 1124 patients with severe TBIs, and 999 total patients were included in the study. This study received approval from the institutional review board (IRB) at Elmhurst Facility with IRB number 24-12-092-05G.

Data for all patients with severe traumatic brain injury was requested from the National Trauma Registry of the American College of Surgeons (NTRACS) database at our center, Elmhurst Hospital Center. NTRACS gathers various categories of TBIs. To maintain clarity in our table and results, we have included only the combinations of TBIs that are relevant to our study. When necessary, we reviewed the patient’s medical charts to collect all pertinent information required for this research. Patients were identified based on the injury mechanism, cause of injury, primary mechanisms (lCD9 or lCDL0 E-Code), and the AIS score (head). The AIS score ranges from 1 to 6 per body region. The breakdown of the broad types of injury with subdural hemorrhage being the most prevalent, followed by subarachnoid, intraparenchymal, other injuries, epidural, and lastly concussions. Severe TBI is further defined by a Glasgow Coma Scale (GCS) score of 8 or less after resuscitation but before sedation.

### 2.2. Data Collection

We collected data using a data collection tool (Excel sheet or spreadsheet). We incorporated all data elements into this tool, for example, patient demographics, clinical outcomes, and biochemical data. Baseline admission data included data elements like demographics (age, sex, race, ethnicity), the AIS, and the International Classification of Diseases (ICD) injury description. Age ranges were divided into groups: pediatric (less than 18), young adults (ages 18 to 45), older adults (ages 46 to 75), and elderly (ages greater than 75). Sex was categorized as male and female.

We also investigated based on the types of injuries—blunt versus penetrating—as well as the mechanisms of injury, which included falls, pedestrians being struck, blunt assaults, motor vehicle collisions (MVCs), minor MVCs, penetrating assaults, and other causes. Additionally, we classified intracranial injuries into broader categories: subdural, subarachnoid, epidural, intraparenchymal injuries, and concussions. Further subset analysis was performed based on the unique intracranial ICD-10 codes assigned to each patient.

The SL was obtained manually from charts at four pre-determined time frames of a patient’s hospital stay. These were hospital admission, ICU admission, ICU discharge, and then either death or hospital discharge. “Admission” was defined as the first measured level of a metabolite after admission to the trauma bay. “ICU admission” was defined as the first measured level of a metabolite after admission to the ICU while “ICU discharge” was the last measured level of a metabolite before arrival in a step-down or floor unit. “Hospital discharge” was the last measured value for a metabolite before discharge and “death” was the last measured value before the time of death.

In a few cases, missing data were filled in by taking the most recent sodium measurement in a given defined time frame, even if that sodium value was used for another time frame. For example, if a patient was discharged with only one set of lab values obtained, the sodium level that was used for admission was also used for discharge as that was technically the first SL obtained during hospital admission and the last obtained before hospital discharge. Given the universal use of basic metabolic panels and blood gasses, the amount of missed sodium values, while not counted, was negligible for our study.

The primary outcome variable in this dataset was mortality. Secondary outcomes were hospital LOS, ICU LOS, and days requiring mechanical ventilation. All secondary outcomes were measured in days and mortality was assigned a binary variable, 0 for no mortality recorded and 1 if a patient was a recorded mortality.

We analyzed data using both Excel and R Studio (Version: 2024.09.1+394). A two-tailed *t*-test and single-factor ANOVA with significance levels of 0.05 were used to examine any potential differences in SL with demographic factors and injury patterns. To convert SL to a categorial variable for these analyses, patients were assigned a sodium range based on accepted levels in the literature. SL was categorized into the following ranges: extreme hyponatremia (less than 120 mEq/L), hyponatremia (120–135 mEq/L), normonatremia (135–145 mEq/L), hypernatremia (145–160 mEq/L), and extreme hypernatremia (>160 mEq/L) [10,11,12]. Primary outcome variables were studied using univariate linear regression models in R while subset analysis was performed on patients with one, two, three, and four or more assigned intracranial ICD-10 codes.

## 3. Results

A total of 999 patients were included in the study after applying the exclusion criteria, which resulted in 125 dropouts. Among these patients, 758 were male and 241 were female. The age distribution was as follows: 26 patients were in the pediatric group, 389 in the younger adult group, 380 in the older adult group, and 203 in the elderly group. The average age of all participants was 52.87 years. Out of the reported injuries, 984 were due to blunt trauma, while only 15 were classified as penetrating trauma. A total of 717 patients had an ICD-10 code classified as a subdural hemorrhage, 439 as a subarachnoid hemorrhage, 104 as epidural hemorrhage, 212 as intraparenchymal hemorrhage, 23 as a concussion, and 173 as other. This means that in our study, on average, each patient averaged approximately 1.82 ICD-10 codes associated with intracranial injuries. Moreover, 90 patients in our study died, and 909 patients survived their index hospitalization. The average length of hospitalization was 11.66 days, the average ICU stay was 3.71 days, and the average length of ventilator was 1.73 days.

There were multiple significant relationships found between SL and various prognostic factors and outcomes. In this study, an analysis of variance (ANOVA) was used to compare variance across means of distinct groups, and a linear regression analysis was used to predict outcomes based on dependent variables.

We found statistically significant results between age range and admission SL and ICU admission (day) SL (*p* = 8.79 × 10^−8^, *p* = 0.000205). Differences in SLs upon admission were statistically significant when compared across four age ranges (pediatric, young adult, older adults, and elderly) (*p* = 2.394 × 10^−6^). ICU admission (*p* = 0.00633), hospital discharge (*p* = 0.0266), and death (*p* = 0.0413) showed a significant difference as well but with higher *p*-values. There was no statistical significance in ICU discharge (*p* = 0.3358). A statistically significant difference was found in average SL during a hospital stay in male versus female and blunt versus penetrating injury. The only exception was a difference in ICU discharge levels when comparing blunt versus penetrating injury (*p* = 1.304 × 10^−8^), but the penetrating injury sample size was small, which might have skewed the results shown in Table 1.

We additionally compared ranges of SLs (extreme hyponatremia, hyponatremia, normonatremia, hypernatremia, extreme hypernatremia) and found statistically significant differences in the sodium ranges at admission, ICU admission, and ICU discharge when compared to admission ISS scores and GCS scores, as seen in Table 2. While the ANOVA could not specifically provide the direction of the correlation, the general trend showed that increases in ISS and decreases in GCS were associated with increases in admission, ICU admission, and ICU discharge SLs.

As a proxy for overall severity of injury, we used ANOVA to compare the SLs at the five time frames for different subtypes of intracranial injuries (subdural hemorrhage, subarachnoid hemorrhage, epidural hemorrhage, intraparenchymal hemorrhage, concussion, and other), as well as for different numbers of intracranial injuries. Across admission, ICU admission, ICU discharge, hospital discharge, and death SLs, there were no statistically significant differences comparing the subtypes of intracranial injuries (*p* = 0.78; *p* = 0.75; *p* = 0.44; *p* = 0.23; *p* = 0.66) or several different coded injuries (*p* = 0.47; *p* = 0.39; *p* = 0.30; *p* = 0.25; *p* = 0.98).

Linear regression analysis revealed a negative correlation between patient age and both admission SL (*p* = 8.79 × 10^−8^; coefficient = −0.035) and ICU admission SL (*p* = 0.000205; coefficient = −0.033). There was additionally a strong statistically significant positive correlation between ICU admission serum SL (SL) and Injury Severity Score (ISS) (*p* = 6.927 × 10^−9^; coefficient = 0.36056), and a negative correlation with GCS (*p* = 1.7525 × 10^−11^; coefficient = −0.21914). Statistically significant differences were observed when comparing ICU admission SL to hospital LOS (*p* = 0.0367; coefficient = 0.3857), ventilator days (*p* = 0.00201; coefficient = 0.19), ICU LOS (*p* = 0.0233; coefficient = 0.1545), and mortality (*p* = 0.000285; coefficient = 0.0096), as seen in Table 3 with additional subgroup analysis including the breakdown of patients by how many different intracranial injuries they were diagnosed with.

Further analysis of the data was performed with subgroups based on the category of intracranial injury as well as several different injuries. In a regression analysis examining the number of types of injuries, ICU admission sodium was positively correlated with ventilator days and mortality in patients with more than three types of injuries. ICU discharge sodium was positively correlated with mortality in patients with two, three, and more than three types of injuries. In patients with single injuries, discharge sodium was positively correlated with mortality in patients with subdural hematomas with *p*-values of 0.0002 (coefficient = 0.0141), respectively. Also, in single injuries, ICU admission sodium was positively correlated with mortality in patients with subdural hemorrhages with *p*-values of 0.0216 (coefficient = 0.0082) and negatively corrected with mortality in patients with subarachnoid hemorrhages with a *p*-value of 0.0342 (coefficient = −0.03395).

Finally, the variation in SLs from hospital admission to ICU admission was positively correlated with hospital LOS (*p* = 0.0148), as shown in Figure 1, and ventilator days (*p* = 0.0261) and mortality (*p* = 6.9 × 10^−8^), as shown in Table 4.

Given the results that increased SL was universally associated with worse outcomes, further analysis was performed on only patients with normal or low SLs (cutoff: 145 mEq/L) to exclude patients with existing metabolic derangements that would be worsened by the administration of hypertonic saline. The same linear regression models again found that the change in SLs from trauma bay admission to ICU admission was positively correlated with hospital LOS (*p* = 0.00168; coefficient = 0.7708), ventilator days (*p* = 0.00295; coefficient = 0.209) and mortality (*p* = 8.3 × 10^−8^; coefficient = 0.0187), showing that even in patients with already low SLs, increases in sodium were associated with worse outcomes. Figure 2 of this paper shows the linear regression model between change in SL from hospital admission to ICU admission and its overall effect on hospital length of stay measured in days whereas Figure 3 illustrates a box plot illustrating change in SL from hospital admission to ICU admission compared to mortality. Additionally, 0 represents no mortality, and 1 represents recorded mortality.

## 4. Discussion

### 4.1. Key Findings

This single-center study aimed to explore the complex relationship between serum SL at different stages of hospitalization and the prognostic outcomes for patients with TBIs. Our analysis revealed that hypernatremia, which is characterized by elevated serum SLs, is significantly associated with poorer outcomes in this vulnerable group. Hypernatremia is a critical factor contributing to morbidity and mortality in patients with neurological disorders. For instance, in individuals with subarachnoid hemorrhage, hypernatremia was linked to increased mortality rates and worse overall outcomes [11]. Additionally, research has shown that hypernatremia in a heterogeneous group of patients treated in intensive care units is associated with higher rates of morbidity [12].

Severe TBI frequently results in complex disturbances in serum SLs and water regulation, complicating patient management. Several factors influence SL, including the use of osmotherapy, the administration of high sodium fluids, and the selection of nutritional preparations—all determined by the treatment provided [2]. The management of SL can be particularly challenging due to the multifactorial nature of TBIs. For instance, patients often present with a combination of cerebral edema, altered fluid balance, and endocrine dysfunction, all of which can influence sodium homeostasis. Additionally, some factors are related to the trauma itself. For example, posterior pituitary dysfunction can occur because of TBIs, leading to a deficiency in the secretion of antidiuretic hormone (ADH). This deficiency results in excessive water loss and, subsequently, hypernatremia. Conversely, the overproduction of ADH can cause water retention, heightening the risk of hyponatremia [2,3].

The exact etiology of hypernatremia in TBI patients is multifaceted and not strictly defined. It may arise from various causes, including hyperosmolar therapy, hypovolemia, insensible free water losses, or a high sodium load from intravenous fluids, nutritional feeds, or medications. This complexity necessitates a careful assessment of each patient’s unique circumstances. Furthermore, hypothalamic dysfunction that does not lead to overt diabetes insipidus may also contribute to hypernatremia following a TBI [12]. Such dysfunction could be a result of direct neuronal injury or secondary to the systemic inflammatory response often seen in TBIs, which further complicates fluid and electrolyte management.

The prevalence of hyponatremia and hypernatremia among TBI patients was reported to range between 15% and 55% [4,5] and 30% and 50% [6,7], respectively. Notably, studies that have employed frequent sodium measurements indicate that as many as 64% of TBI patients experience hypernatremia [13]. This alarmingly high prevalence underscores the clinical relevance of our findings and emphasizes the urgent need for increased awareness among healthcare providers regarding the implications of SLs in the management of TBIs. Continuous monitoring and the early detection of sodium imbalances may be crucial for improving patient outcomes, as even mild disturbances can significantly affect neurological recovery and overall health.

Our study revealed a strong correlation between SLs recorded at the time of ICU admission and both the ISS and GCS. These two metrics are well-established indicators of injury severity and prognosis in TBI patients. While a lack of knowledge of potential confounding factors means that we cannot say SL independently is a good prognostic indicator, it is a much more readily available metric that we now know is in line with better prognostic values. Our findings support the notion that a higher SL upon ICU admission is associated with greater injury severity and poorer neurological function and persistently elevated SL. Moreover, sodium levels and increases in SLs throughout the early stages of a patient’s admission are associated with worse outcomes. This relationship is particularly important as it establishes SL as a potential predictive marker for adverse outcomes in TBI patients, aligning with the previous literature that emphasizes the significance of early interventions based on initial lab results. Clinicians may need to consider SL alongside traditional scoring systems when assessing patient prognosis and determining treatment plans.

Increases in SLs from admission to the trauma bay and then to the ICU are associated with longer hospital stays, extended periods of mechanical ventilation, and higher mortality rates among patients. These associations occur within a critical time frame during which trauma surgeons have the opportunity to intervene and potentially change the course of a patient’s clinical outcome. By understanding the implications of fluctuations in SLs, healthcare professionals can gain valuable insights that may significantly influence treatment decisions and patient management strategies in clinical settings.

The observed relationship between SL fluctuations and adverse outcomes strongly suggests that targeted interventions, implemented early—such as in the emergency room—and aimed at normalizing SLs, could lead to improved prognoses for patients with TBIs. For instance, implementing targeted therapies designed to effectively manage conditions such as hypernatremia or hyponatremia could optimize recovery trajectories for TBI patients. This approach has the potential not only to enhance individual patient outcomes but also to alleviate the overall burden on healthcare systems by reducing complications and associated healthcare costs.

By identifying these important relationships, our study significantly enhances the understanding of controlling/handling SLs in patients with TBIs and how this knowledge can lead to improved clinical practices. This insight may contribute to the development of strategies aimed at enhancing patient outcomes. Given the complexities involved in fluid and electrolyte management for TBI patients, there is a strong need for further research. Future studies should specifically focus on creating and implementing standardized protocols for sodium monitoring and intervention strategies to improve patient survival rates.

### 4.2. Implications of Study’s Findings

The clinical implications of our study’s findings present a significant challenge to the traditional paradigms that currently guide the management of TBIs, particularly regarding the administration of hypertonic saline. Historically, hypertonic saline was widely regarded as a beneficial treatment option for reducing intracranial pressure, which is a crucial factor in the management of TBIs. This treatment was integrated into clinical practice based on its perceived ability to effectively manage cerebral edema and improve patient outcomes. However, our findings, in conjunction with emerging research in the field, suggest that it may be time for a re-evaluation of this long-standing approach. While hyperosmolar therapy can provide certain benefits in treating cerebral edema, the potential morbidity associated with hypernatremia—an elevation of SL in the blood—must be carefully weighed against these advantages. This balance necessitates frequent monitoring of serum SLs during hyperosmolar therapy to prevent rapid and substantial increases in serum sodium, which can lead to serious complications [14].

Furthermore, the landscape of TBI management is evolving, and there is a growing recognition of the complexity inherent in treating these patients. Recent studies have indicated that patients with TBI who received hypertonic saline did not demonstrate a statistically significant improvement in their likelihood of achieving favorable outcomes at discharge or even at the six-month follow-up when compared to those who received standard crystalloid solutions [14]. This finding is particularly noteworthy as it suggests that hypertonic saline may not effectively reduce mortality rates; however, it is important to acknowledge that treatment with hypertonic saline did lead to a shorter LOS in the ICU [14]. While a reduced LOS can be beneficial from a healthcare resource perspective, it raises questions about the overall efficacy of hypertonic saline in improving long-term patient outcomes.

Our data further proposes that a narrow therapeutic window exists for clinical interventions aimed at enhancing outcomes in critically ill patients with TBIs. The observation that a higher SL at the time of ICU admission, along with significant increases in SLs during hospitalization, correlates with adverse outcomes presents a critical opportunity for targeted interventions. This insight emphasizes the need for a more nuanced approach to managing SLs in TBI patients, especially early in the hospital course, advocating for treatment strategies tailored to individual patient responses and clinical conditions, such as expedited lab draws in emergency rooms. It is essential to recognize that TBI patients are not a homogeneous group; their clinical profiles can vary widely based on factors such as age, severity of injury, comorbid conditions, and individual responses to treatment.

This variability raises several vital questions for clinical practice. For example, could therapies specifically designed to lower SLs during the critical period leading up to ICU admission significantly improve patient outcomes? Our results suggest that such interventions could lead to reductions in hospital LOS, decreased reliance on mechanical ventilation, and lower overall mortality rates. This possibility underscores the necessity for ongoing dialog and research concerning management protocols for TBI patients, as re-evaluating sodium management could play a key role in enhancing patient-centered outcomes.

Moreover, understanding sodium’s role extends beyond the mere measurement of its levels; it encompasses a broader physiological context in which electrolyte balance can profoundly affect neurological outcomes. The intricate interplay between SL and brain function indicates that the careful monitoring and management of serum sodium may not only influence immediate clinical outcomes but also impact long-term recovery trajectories for patients with TBIs [12]. For instance, dysregulated SLs can lead to neurological dysfunction, affecting cognitive recovery and overall rehabilitation success. This underscores the importance of considering SL as an integral component of a comprehensive approach to TBI management, which should also incorporate both medical and rehabilitative strategies to optimize recovery [12].

In conclusion, our study emphasizes the urgent need to re-evaluate current SL management protocols for patients with TBIs. This re-evaluation is crucial for improving neurological health and patient outcomes. By developing a better understanding of sodium’s impact, we can create innovative, evidence-based treatment strategies that enhance the quality of care.

Collaboration among clinicians, researchers, and healthcare systems is vital for establishing standardized protocols for sodium monitoring and intervention. Future research should focus on multicenter studies to validate our findings. This comprehensive approach aims not only to improve immediate survival but also to prioritize long-term recovery and quality of life for individuals with TBIs, ultimately enhancing clinical outcomes and patient care.

### 4.3. Strengths and Limitations

Our study provides valuable insights into the relationship between SLs and outcomes in patients with TBIs. However, we must recognize some limitations in our research design that could influence how our findings are interpreted. Since this is a single-center study, the external validity of our results may be limited, which means they might not apply to different healthcare settings and populations. This limitation is particularly significant in the context of TBIs, as various institutions may use different protocols, treatment approaches, and patient management strategies. Consequently, the specific practices and outcomes observed in our study may not fully reflect those found in other clinical environments, which could impact the applicability of our findings. Single injury patterns (i.e., subarachnoid, epidural, intraparenchymal, concussion, and other TBI groups), due to their small sample sizes, did not allow for regression analysis without yielding erroneous conclusions.

Additionally, our study was conducted retrospectively, which introduces its own set of limitations. Retrospective studies often face potential biases related to data collection and interpretation, particularly concerning the accuracy and completeness of medical records. However, it is important to note that SLs were objectively collected and consistently measured during patient hospitalization, allowing for a reliable analysis of these critical data points. We examined several important prognostic outcomes, such as mortality rates, hospital LOS, ICU LOS, and the number of days on mechanical ventilation among TBI patients. This comprehensive assessment helps to minimize concerns regarding selection bias and ascertainment bias, lending greater credibility to our findings.

A unique aspect of our study is its consideration of both ISS and GCS as measures for assessing mortality. These metrics provide robust patient-centered outcome measures that enhance the validity of our findings. However, it is also important to recognize that our study did not include direct measurements of neurological outcomes for the patients. Such assessments would require longer follow-up periods, potentially extending from six months to one year after hospital discharge. This limitation highlights the need for future research to explore the long-term implications of SLs on neurological recovery and overall quality of life for TBI patients. Longitudinal studies could offer valuable insights into how variations in sodium management impact recovery trajectories over time.

Another limitation of this study was the incomplete data regarding the hospital courses of the patients. There were a small number of missing sodium values, which were not optimally collected, as they did not occur at the exact time of ICU or hospital discharge. More importantly, there was a lack of data about the patient’s hospital courses, making it difficult to obtain a complete clinical picture. This limitation hinders our ability to assess how interventions and complications during the hospital stay influenced the measured SLs. However, the most crucial SLs were collected during the admission and ICU admission time frames, which were less likely to be affected by the hospital course since they were typically recorded within the first few hours after the patient arrived in the trauma bay.

Furthermore, as our study was not designed as a randomized controlled trial, we must exercise caution when concluding causality. While our findings suggest significant associations between SLs and various patient outcomes, definitive causal relationships cannot be established based solely on this retrospective analysis. Correlation does not imply causation, and the relationships observed may be influenced by confounding variables not accounted for in our study. Future research, particularly prospective and randomized controlled trials, will be crucial to determine the directionality and underlying mechanisms of these associations. Such studies could help clarify whether interventions aimed at normalizing SLs can lead to improved outcomes for TBI patients.

Given our findings, the importance of frequent monitoring of SLs in TBI patients cannot be overstated. Rather than studying SLs and changes at three specific points during hospitalization, a more dynamic approach involving frequent checks—such as every six hours—would provide more relevant data to study the impact of sodium level changes on patient outcomes. In practice, more frequent sodium checks allow for timely adjustments to therapies based on the patient’s fluctuating SLs, helping to prevent complications such as hypernatremia and hypovolemia. Routine assessments can facilitate proactive management strategies that enhance patient safety and improve overall outcomes.

In summary, while our study contributes valuable knowledge to the field of TBI management, the limitations inherent in our research design warrant careful consideration. By acknowledging these limitations, we can better understand the context of our findings and highlight the need for further investigation. Future studies should aim to address these gaps by employing larger, multicenter cohorts and utilizing rigorous methodologies that can strengthen the evidence base for sodium management in TBI patients. Ultimately, addressing these limitations will be essential to advancing our understanding and improving clinical practices related to SLs in this vulnerable population.

### 4.4. Future Perspectives

More research is needed to investigate the complex relationship between SL and patient-centered outcomes in TBIs. While our study identifies significant links between sodium fluctuations and negative outcomes, important questions remain. It is essential to conduct prospective studies involving larger and more diverse populations to validate these findings and to better understand how demographic factors and comorbidities affect SLs and recovery.

Targeted interventions to modulate SLs, such as fluid management, dietary adjustments, and pharmacological strategies, warrant investigation to identify effective protocols. Multicenter studies could enhance the understanding of TBI pathophysiology and improve generalizability through standardized sodium management practices. Additionally, longitudinal research should assess the long-term effects of SL management on neurological outcomes and quality of life. This broader perspective can provide critical insights into recovery trajectories beyond immediate survival rates.

In conclusion, SLs should be considered a critical indicator in TBI management, guiding dynamic treatment strategies. Continued research and multi-site collaboration will refine care approaches, aiming to improve outcomes for individuals with TBIs.

## 5. Conclusions

This study from a single level-1 trauma center has shown a significant link between hypernatremia and poorer outcomes in patients with TBIs. In trauma care, the roles of the trauma team and emergency room doctors are essential, as they are often the first to assess and manage patients upon arrival. This requires them to make rapid decisions. These teams collaborate closely with neurosurgeons to carry out critical interventions such as fluid resuscitation and medication administration. Effective communication and collaboration are crucial for optimizing patient outcomes, especially for those with severe TBIs.

Once a patient is admitted, a multidisciplinary approach that includes trauma physicians, neurosurgeons, intensivists, nursing staff, and rehabilitation specialists is vital. This team works together to develop individualized treatment plans that address the complex factors affecting patient care and survival. Ultimately, this approach leads to improved monitoring and interventions for patients [12].

Our findings indicate that SL upon ICU admission is correlated with both ISS and GCS. Specifically, an elevated SL at admission was linked to adverse hospital outcomes, including prolonged LOS, extended ICU stays, increased days of mechanical ventilation, and higher mortality rates. This has clinical implications as this study challenges the traditional use of hypertonic saline for reducing intracranial pressure in TBI patients, as the potential risks associated with its use must be weighed against the benefits. Moreover, our results suggest that variability in serum SL is independently associated with mortality throughout the hospital stay, irrespective of the absolute serum sodium concentration. This highlights the importance of not only the sodium level itself but also its fluctuations during treatment. Further prospective investigations are essential to confirm the clinical significance of both SLs and their variability in larger cohorts of TBI patients.

There is a critical need for a nuanced, patient-specific approach to managing serum SLs in patients with TBIs. This approach should emphasize early and frequent monitoring of serum SL. Additionally, investigating whether targeted clinical interventions aimed at lowering serum sodium levels before ICU admission and reducing sodium variability can lead to improved patient outcomes would be beneficial. Although there is a clear correlation between fluctuations in sodium levels and adverse outcomes, it remains uncertain whether this correlation is a direct cause of those outcomes or simply a secondary indicator of disease severity.

If there are other indirect factors associated with this issue, it would be worthwhile to explore them in future studies. Additionally, investigating whether interventions designed to reduce SL variability can improve patient outcomes would be valuable. Conducting a multicenter study would offer a more comprehensive understanding of the complex pathophysiology related to TBIs and help validate our findings across diverse patient populations. This broader perspective could inform clinical practices and enhance care for this vulnerable group of patients.

## Figures and Tables

**Figure 1 diagnostics-15-00125-f001:**
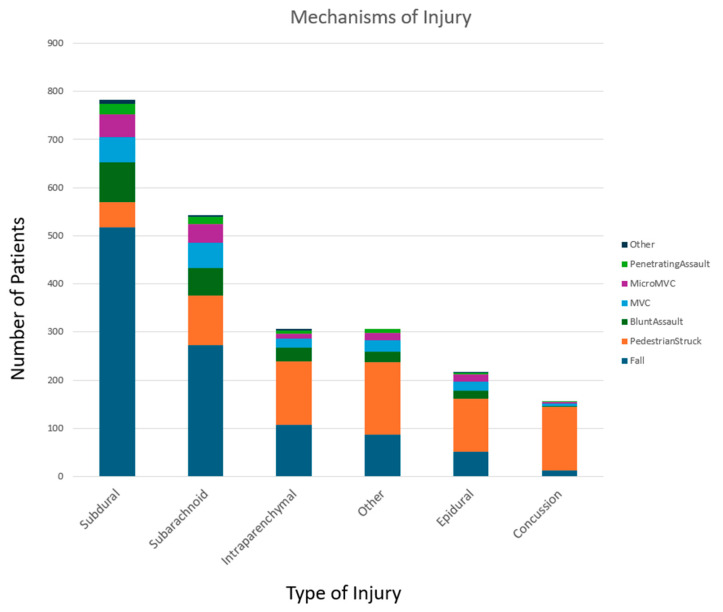
Most common mechanisms associated with different patterns of traumatic brain injury (TBI).

**Figure 2 diagnostics-15-00125-f002:**
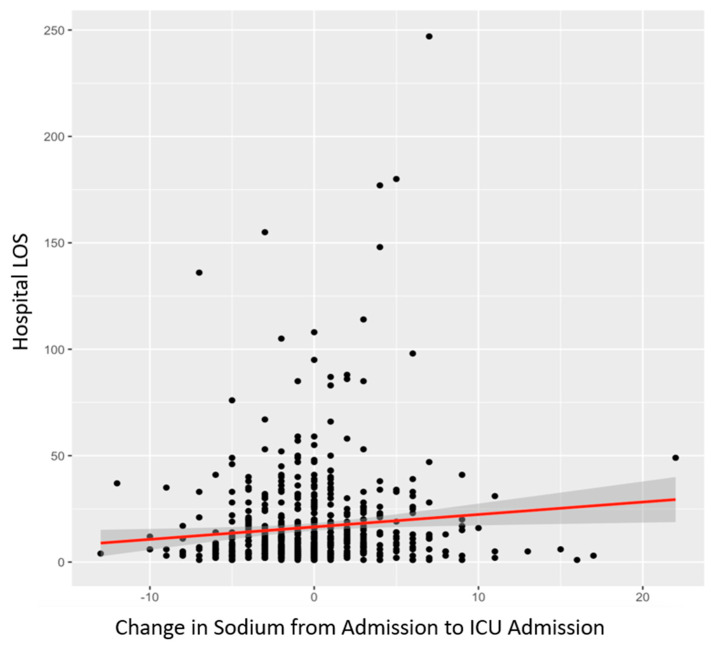
Linear regression model between change in SL from hospital admission to ICU admission and its overall effect on hospital length of stay measured in days. Change in SL measured in mEq/L.

**Figure 3 diagnostics-15-00125-f003:**
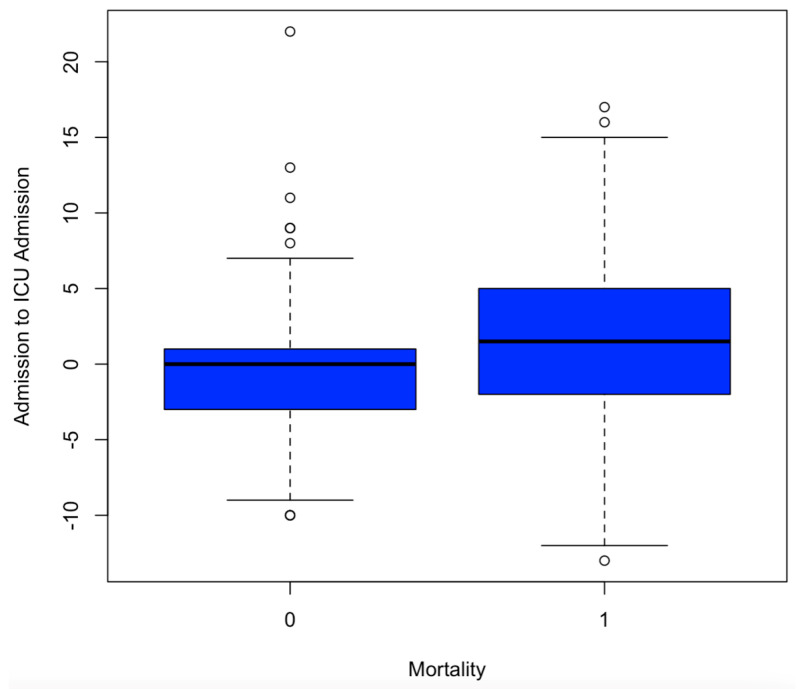
Box plot illustrating change in SL from hospital admission to ICU admission compared to mortality. Additionally, 0 represents no mortality, and 1 represents recorded mortality. Units are in mEq/L, a standard measuring unit for serum SL.

**Table 1 diagnostics-15-00125-t001:** ANOVA analysis comparing SL at five time frames against demographic factors and injury factors.

		Admission	ICU Admission	ICU Discharge	Hospital Discharge	Death
**Sex**	Female	138.929461	138.618705	138.6834532	138.8340807	142
	Male	139.5	139.3747454	138.7800407	138.3846154	144.541667
	*p*-value	0.09075535	0.109418455	0.884626824	0.094201029	0.18000407
**Age Range**	Under 18	139.346154	138.6363636	138.8181818	138.0769231	
	18–45	140.249357	139.9266409	138.2007722	138.6280992	144.851852
	46–74	139.05	139.0254237	138.970339	138.1196581	146.034483
	75+	138.254902	138.1048387	139.516129	139.0406977	141.676471
	*p*-value	2.3935 × 10^−6^	0.00633139	0.335778331	0.0266175	0.04134098
**Injury Type**	Blunt	139.361789	139.2125604	138.9452496	138.4933333	144.045977
	Penetrating	139.4	138.8888889	125.8888889	138.5833333	143.666667
	*p*-value	0.97433071	0.84467158	1.30 × 10^−8^	0.929255109	0.92881589
**Injury Mechanism**	Fall	138.89	138.77	138.77	138.43	144.09
	Blunt Assault	140.19	139.88	138.85	138.84	141
	MVC	140.47	139.88	138.97	138.52	145.33
	Pedestrian Struck	139.76	140.22	139.8	138.81	141.67
	Micro MVC	140.03	138.83	139.88	138.21	147.2
	Penetrating Assault	139.77	140.07	140.27	138.5	145.5
	Other	139.5	142.67	134.83	137.6	
	*p*-value	0.005566	0.07156	0.159	0.8247	0.7797
**Diagnosis**	Subdural	139.295676	139.2985386	139.0041754	138.0911271	145
	Subarachnoid	139.410023	139.4181818	139.3090909	138.2367021	143.984127
	Epidural	139.980769	139.1318681	138.1978022	137.8969072	141
	Intraparenchymal	139.216981	138.8387097	139.1806452	138.0549451	143.4
	Concussion	139.73913	138.8888889	142.4444444	138.9545455	153.00
	Other	139.271676	139.7323944	139.2535211	138.8321678	143.9
	*p*-value	0.78006053	0.746736793	0.443340882	0.232330769	0.65649296
**Number of Injuries**	One	139.442177	139.1829787	138.8212766	138.6510539	144.375
	Two	138.990164	138.8550725	138.7777778	138.2919708	144.0625
	Three	139.495935	139.6153846	139.0288462	137.970297	144.272727
	Four+	139.712329	139.8714286	140	138.20	143.5
	*p*-value	0.46740898	0.386629866	0.303727483	0.251310921	0.98487618

**Table 2 diagnostics-15-00125-t002:** Two-tailed *t*-test comparing sodium ranges with ISS and GCS scores.

		Extreme Hyponatremia	Hyponatremia	Normonatremia	Hypernatremia	Extreme Hypernatremia	*p*-Value
**Admission**	ISS	15	18.3508772	17.5207592	20.8684211	25	0.04526659
	GCS	14.6666667	13.25	13.2846803	11.2777778	6	0.00366251
**ICU Admission**	ISS	9	19.7848101	19.3333333	27.6818182	30	6.9269 × 10^−9^
	GCS	15	12.8333333	12.7586912	8.38095238	5	1.7248 × 10^−11^
**ICU Discharge**	ISS	14	20.275	19.2785571	26.8541667	35	5.5463 × 10^−8^
	GCS	15	12.6202532	12.6419753	10.0681818	3	0.00017872
**Discharge**	ISS		18.34	16.5708861	16.8181818		0.07480594
	GCS		13.4343434	13.7005141	13.1363636		0.50224051
**Death**	ISS		23	27.3695652	28.5	35	0.59882384
	GCS		7.28571429	8.86666667	8.3125	3	0.7079669

Note: Extreme hyponatremia is less than 120 mEq/L, hyponatremia is 120–135 mEq/L, normonatremia is 135–145 mEq/L, hypernatremia is 145–160 mEq/L, and extreme hypernatremia is >160 mEq/L [10].

**Table 3 diagnostics-15-00125-t003:** Linear regression analysis comparing SL at three time frames (at admission, ICU admission, and ICU discharge) against outcome variables including subset analysis based on many diagnosed injuries (hospital LOS, ICU LOS, ventilator days, and mortality).

	Time Frame	Outcome	*p*-Value	Coefficient
**Overall**	**Admission**	Hospital LOS	0.854	2.48 × 10^−2^
		ICU LOS	0.105	0.08177
		Ventilator Days	0.105	0.06964
		Mortality	0.688	−0.0008001
	**ICU Admission**	Hospital LOS	0.0367	0.3857
		ICU LOS	0.0233	0.1545
		Ventilator Days	0.00201	0.18987
		Mortality	0.000285	0.009556
	**ICU Discharge**	Mortality	3.74 × 10^−10^	0.011598
**One Diagnosed Injury**	**Admission**	Hospital LOS	0.539	0.08503
		ICU LOS	0.81	0.01151
		Ventilator Days	0.504	0.02189
		Mortality	0.737	−0.0006537
	**ICU Admission**	Hospital LOS	0.3026	−0.1744
		ICU LOS	0.761	−0.02321
		Ventilator Days	0.976	−0.001719
		Mortality	0.695	0.001224
	**ICU Discharge**	Mortality	0.0413	0.003263
**Two Diagnosed Injuries**	**Admission**	Hospital LOS	0.576	0.1469
		ICU LOS	0.411	0.08011
		Ventilator Days	0.873	0.009245
		Mortality	0.202	−0.004704
	**ICU Admission**	Hospital LOS	0.0681	0.7054
		ICU LOS	0.206	0.1804
		Ventilator Days	0.0771	0.15391
		Mortality	0.137	0.007462
	**ICU Discharge**	Mortality	2.45 × 10^−6^	0.023185
**Three Diagnosed Injuries**	**Admission**	Hospital LOS	0.36	0.4726
		ICU LOS	0.0792	0.3637
		Ventilator Days	0.303	0.1693
		Mortality	0.956	0.0004212
	**ICU Admission**	Hospital LOS	0.264	0.537
		ICU LOS	0.256	0.2189
		Ventilator Days	0.304	0.1601
		Mortality	0.179	0.009299
	**ICU Discharge**	Mortality	2.01 × 10^−7^	0.037599
**Four or More Diagnosed Injuries**	**Admission**	Hospital LOS	0.324	−0.8928
		ICU LOS	0.955	0.01226
		Ventilator Days	0.262	0.4462
		Mortality	0.259	0.01338
	**ICU Admission**	Hospital LOS	0.721	0.2754
		ICU LOS	0.114	0.2875
		Ventilator Days	0.0482	0.665
		Mortality	0.000871	0.03112
	**ICU Discharge**	Mortality	0.00372	0.024522

**Table 4 diagnostics-15-00125-t004:** Linear regression analysis comparing changes in SLs (changes in SLs from hospital admission to ICU admission, as well as changes from ICU admission to discharge) to outcome variables (hospital LOS, ICU LOS, ventilator days, and mortality).

Time Frame	Outcome	*p*-Value	Coefficient
**Admission to ICU Admission**	Hospital LOS	0.0148	0.5845
	ICU LOS	0.223	0.10799
	Ventilator Days	0.0261	0.17813
	Mortality	6.91 × 10^−8^	0.018351
**ICU Admission to ICU Discharge**	Hospital LOS	0.481	−0.08866
	ICU LOS	0.784	−0.01269
	Ventilator Days	0.571	0.02379
	Mortality	1.10 × 10^−5^	0.010795

## Data Availability

The data were requested from the Elmhurst trauma registry and extracted using electronic medical records after receiving approval from the institutional review board at our facility (Elmhurst Hospital Center).

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
