# Peer review of "Natremia Significantly Influences the Clinical Outcomes in Patients with Severe Traumatic Brain Injury"

_diagnostics, 2025, doi:10.3390/diagnostics15020125_

Round 1

Reviewer 1 Report (Previous Reviewer 1)

Comments and Suggestions for Authors

The authors have made suggested changes.

Author Response

Greetings,

Thank you so much for providing comments "The authors have made suggested changes".

Best regards,

Bharti

Reviewer 2 Report (New Reviewer)

Comments and Suggestions for Authors

·      The monocentric nature of the study limits the generalizability of its findings to other institutions with different protocols. It would be beneficial to conduct a multicentric study in order to confirm the findings across different settings. 

·      Improve the statistical robustness of smaller subgroups, such as patients with penetrating injuries, as the limited number of cases reduces the statistical power of analyses for this subgroup.

·      The retrospective design, although suitable for analyzing historical data, restricts the ability to control for potential confounding factors.

·      The study does not evaluate two crucial elements for a comprehensive clinical assessment: functional recovery and quality of life after discharge. It would be beneficial to include the follow-up to assess neurological recovery, functional outcomes and quality of life after hospital discharge and investigate whether variability in sodium levels correlates with cognitive and rehabilitative trajectories. 

·      Translating findings into practical strategies, e.g. optimising hypertonic saline use and adjusting fluid therapy, must be emphasised. 

·      While there is a demonstrable correlation between variability in sodium levels and adverse outcomes it remains unclear whether this is a direct cause or a secondary marker of disease severity.

·      Although p-values are provided, the presentation of results could be enhanced by incorporating confidence intervals and effect sizes, offering a more thorough and meaningful interpretation of the findings.

Comments on the Quality of English Language

English form needs minor revision.

Author Response

Dear reviewer,

Thank you so much for providing these comments. We have tried to respond to all of them. Please find point to point responses below:

  1. English form needs minor revision.

Response: We have revised entire manuscript and made changes (wherever applicable)

Other comments (can be improved)

  1. The monocentric nature of the study limits the generalizability of its findings to other institutions with different protocols. It would be beneficial to conduct a multicentric study in order to confirm the findings across different settings. 

Response: We appreciate the idea of multisite exploration but before conducting a multisite study, we decided to study the effect of Natremia in patients with severe TBI at our site. According to our research team, conducting a single site study is a thoughtful approach towards a large-scale study design. Also, we have mentioned it in implications of study findings, future perspectives as well as limitations.

Line # 397-400 Collaborative efforts among clinicians, researchers, and healthcare systems are essential for establishing standardized protocols for sodium monitoring and intervention strategies, with future research focusing on multicenter studies to validate our findings

Line # 407-411 Being a single-center study, the external validity of our results may be restricted, not allowing for generalizability across diverse healthcare settings and populations. This limitation is particularly important in the context of TBI, as different institutions may employ varying protocols, treatment approaches, and patient management strategies.

Line # 487-488 Continued research, multi-site collaboration will refine care approaches, aiming to improve outcomes for individuals with TBI.

  1. Improve the statistical robustness of smaller subgroups, such as patients with penetrating injuries, as the limited number of cases reduces the statistical power of analyses for this subgroup.

 Response: We agree and to kindly inform you, we have considered and performed analysis of the data with subgroups based on category of intracranial in-jury as well as a number of different injuries. In a regression analysis examining the number of types of injuries, ICU admission sodium was positively correlated with ventilator days and mortality in patients with more than three types of injuries. ICU discharge sodium was positively correlated with mortality in patients with two, three, and more than three types of injuries. In patients with single injuries, discharge sodium was positively correlated with mortality in patients with subdural hematomas. Also, in single injuries, ICU admission sodium was positively correlated with mortality in patients with subdural hemorrhages and negatively corrected with mortality in patients with subarachnoid hemorrhages.

As a proxy for injury overall severity of injury, we used ANOVA to compare SL at the five timeframes for different subtypes of intracranial injuries (subdural hemorrhage, sub-arachnoid hemorrhage, epidural hemorrhage, intraparenchymal hemorrhage, concussion and other) as well as for different numbers of intracranial injuries. Across admission, ICU admission, ICU discharge, hospital discharge and death SLs, there were no statistically significant differences comparing subtypes of intracranial injuries (p=0.78, p=0.75, p=0.44, p=0.23, p=0.66) or number of different coded injuries (p=0.47, p=0.39, p=0.30, p=0.25, p=0.98).

  1. The retrospective design, although suitable for analyzing historical data, restricts the ability to control for potential confounding factors.

   Response:  Yes, we agree and it has been mentioned in the limitation section.

  1. The study does not evaluate two crucial elements for a comprehensive clinical assessment: functional recovery and quality of life after discharge. It would be beneficial to include the follow-up to assess neurological recovery, functional outcomes, and quality of life after hospital discharge and investigate whether variability in sodium levels correlates with cognitive and rehabilitative trajectories. 

 Response: Discharge dispositions (DD) and GCS scores at the time of discharge are two crucial elements for a comprehensive clinical assessment regarding functional recovery and quality of life after discharge. We have used these both elements but didn’t find any statistical significance worth reporting. Based on DD either the patient was discharged to home/facility/rehab center or other. In the worst-case scenario, the patient died and we used these readings for evaluation as mentioned in our manuscript. At the time of discharge, GCS scores were normal; hence, there was no further need to compare/ evaluate/ study sodium levels with these GCS scores.

  1. Translating findings into practical strategies, e.g. optimising hypertonic saline use and adjusting fluid therapy, must be emphasised. 

Response: Yes, we agree and we have incorporated emphasizing points in our manuscript.

Line # 341-364: The clinical implications of our study's findings present a significant challenge to the traditional paradigms that currently guide the management of TBI, particularly regarding the administration of hypertonic saline. Historically, hypertonic saline has been widely regarded as a beneficial treatment option for reducing intracranial pressure, which is a crucial factor in the management of TBI. This treatment has been integrated into clinical practice based on its perceived ability to effectively manage cerebral edema and improve patient outcomes. However, our findings, in conjunction with emerging research in the field, suggest that it may be time for a reevaluation of this long-standing approach. While hyperosmolar therapy can provide certain benefits in treating cerebral edema, the potential morbidity associated with hypernatremia—an elevation of SL in the blood—must be carefully weighed against these advantages. This balance necessitates frequent monitoring of serum SL during hyperosmolar therapy to prevent rapid and substantial increases in serum sodium, which can lead to serious complications14.

Furthermore, the landscape of TBI management is evolving, and there is a growing recognition of the complexity inherent in treating these patients. Recent studies have indicated that patients with TBI who received hypertonic saline did not demonstrate a statistically significant improvement in their likelihood of achieving favorable outcomes at discharge or even at the six-month follow-up when compared to those who received standard crystalloid solutions14. This finding is particularly noteworthy as it suggests that hypertonic saline may not effectively reduce mortality rates; however, it is important to acknowledge that treatment with hypertonic saline did lead to a shorter LOS in the ICU14. While a reduced LOS can be beneficial from a healthcare resource perspective, it raises questions about the overall efficacy of hypertonic saline in improving long-term patient outcomes.

  1. While there is a demonstrable correlation between variability in sodium levels and adverse outcomes it remains unclear whether this is a direct cause or a secondary marker of disease severity.

Response: It’s a very interesting point. Based on the current study, there is a significant association between hypernatremia and poorer outcomes in patients with TBI and somehow seems to be directly related. Our findings indicate that SL upon ICU admission is correlated with both the ISS and the GCS. Specifically, elevated SL at admission was linked to adverse hospital outcomes, including prolonged lengths of stay (LOS), extended ICU stays, increased days of mechanical ventilation, and higher mortality rates. If there are some other indirect associated factors, those are worth exploring in future studies. We have added it in our future perspectives (line # 521-524).

  1. Although p-values are provided, the presentation of results could be enhanced by incorporating confidence intervals and effect sizes, offering a more thorough and meaningful interpretation of the findings.

Response: Thanks for the suggestion. We have performed all possible analysis for example, a two-tailed t-test comparing sodium ranges with ISS and GCS scores, ANOVA analysis comparing SL at 5-time frames against demographic factors and injury factors, the linear regression analysis comparing SL at three-time frames (at admission, ICU admission, and ICU discharge) against outcome variables including subset analysis based on many diagnosed injuries (Hospital LOS, ICU LOS, Ventilator Days, and Mortality. We have used coefficients along with their P-values. As an example, in the paragraph below, there are numerous values presented. To enhance clarity for the readers, we have focused on providing only the relevant information. This approach will help avoid unnecessary complexity or confusion. We will keep this in mind when designing future manuscripts.

 “Linear regression analysis revealed a negative correlation between patient age and both admission SL (p=8.79x10-8, coefficient=-0.035) and ICU admission SL (p=0.000205, coefficient=-0.033). There was additionally a strong statistically significant positive correlation between ICU admission serum SL (SL) and Injury Severity Score (ISS) (p=6.927 X10-9, coefficient=0.36056), and a negative correlation with GCS (p=1.7525x10-11, coefficient=-0.21914). Statistically significant differences were observed when comparing ICU admission SL to hospital LOS (p = 0.0367, coefficient = 0.3857), ventilator days (p = 0.00201, coefficient = 0.19), ICU LOS (p=0.0233, coefficient = 0.1545), and mortality (p = 0.000285, coefficient = 0.0096) as seen in Table 3 with additional subgroup analysis including breakdown of patients by how many different intracranial injuries they were diagnosed with.” 

We hope you find our manuscript suitable for publication and look forward to hearing from you.

Sincerely,

Bharti Sharma (Corresponding author)

This manuscript is a resubmission of an earlier submission. The following is a list of the peer review reports and author responses from that submission.

Round 1

Reviewer 1 Report

Comments and Suggestions for Authors

Overall, an interesting study with a good sample size showing the relationship between sodium levels and outcomes in patients with severe TBI and its potential prognostic value. 

Please add details what is SL collected

Please add details about analyses (e.g., comparing SL across injury types or comparing the effect of age on outcomes)

Please add details how the missing data were handled.

Please add details for the thresholds used for significance in multiple comparisons 

Please add details for injury patterns (subdural hematoma vs. subarachnoid, etc.) and associated endocrine dysfunctions.

Tables & Figures are well presented.

Please add limitations

Retrospective Design

Missing Data

Generalizability of findings

Small Sample Sizes in Subgroups

Please add a concluding statement discussing the clinical implications of the findings 

Some sections are too long

Comments on the Quality of English Language

Minor editing

Reviewer 2 Report

Comments and Suggestions for Authors

Dear sir,

I have the following comments regarding your article.

Abstract

Needs lucid information regarding sodium status at admission & during hospital stay. Its relationship with disease severity and outcome.

Other major comments

Dysnatremia: is it possible to convert to sodium dysregulation?

Introduction needs to be rewritten emphasising the incidence and its relevance in the management and outcome.

Hypothesis and objectives should be clearer.

What is the incidence pf sodium dysregulation in TBI including both hypo and hypernatremia?

This information also needed in authors’ results. How may many patients have shifted from hypo to hyper and vice versa during treatment.

Sodium level may be altered during the ICU stay rather than at admission and at discharge from ICU. Why did you not considered lowest and the highest values for hypo or hypernatremia?

Have you excluded iatrogenic or pre-existing endocrinal causes?

You may define outcomes as primary and secondary.

Please use conventional terminology for hypo and hypernatremia. There are standard definition.

Results needs reformatting of information with above mentioned comments and reanalysis

Comments on the Quality of English Language

Dear sir,

I have the following comments regarding your article.

Abstract

Needs lucid information regarding sodium status at admission & during hospital stay. Its relationship with disease severity and outcome.

Other major comments

Dysnatremia: is it possible to convert to sodium dysregulation?

Introduction needs to be rewritten emphasising the incidence and its relevance in the management and outcome.

Hypothesis and objectives should be clearer.

What is the incidence pf sodium dysregulation in TBI including both hypo and hypernatremia?

This information also needed in authors’ results. How may many patients have shifted from hypo to hyper and vice versa during treatment.

Sodium level may be altered during the ICU stay rather than at admission and at discharge from ICU. Why did you not considered lowest and the highest values for hypo or hypernatremia?

Have you excluded iatrogenic or pre-existing endocrinal causes?

You may define outcomes as primary and secondary.

Please use conventional terminology for hypo and hypernatremia. There are standard definition.

Results needs reformatting of information with above mentioned comments and reanalysis
